# Anthropocenic Limitations to Climate Engineering

**Jeroen Oomen**

Urban Futures Studio, Faculty of Geosciences, Utrecht University, Heidelberglaan 8, P.O. Box 80125,
NL-3508TC Utrecht, The Netherlands; j.j.oomen@uu.nl

**Abstract:** The development of climate engineering research has historically depended on mostly western, holistic perceptions of climate and climate change. Determinations of climate and climate change as a global system have played a defining role in the development of climate engineering. As a result, climate engineering research in general, and solar radiation management (SRM) in particular, is primarily engaged in research of quantified, whole-Earth solutions. I argue that in the potential act of solar radiation management, a view of climate change that relies on the holistic western science of the climatic system is enshrined. This view, dependent on a deliberative intentionality that seems connected to anthropocenic notions of responsibility and control, profoundly influences the assumptions and research methods connected to climate engineering. While this may not necessarily be to the detriment of climate engineering proposals—in fact, it may be the only workable conception of SRM—it is a conceptual limit to the enterprise that has to be acknowledged. Additionally, in terms of governance, reliability, and cultural acceptance, this limit could be a fundamental objection to future experimentation (or implementation).

**Keywords:** climate engineering; geoengineering; Anthropocene; responsibility; limits

## 1. Introduction

On 1 June 2017, the president of the United States announced that the United States would withdraw from United Nations Framework Convention on Climate Change Paris Agreement. While his decision was met with a fierce backlash—according to *The Economist*, 51% percent of Republican voters and 86% of Democratic voters were opposed to leaving the Paris Accord, and a large number of cities and states announced they would adhere to the Paris accord on their own capacity (The Economist 2017)—it served to underscore, yet again, the precariousness of climate politics. Nominally, there is widespread agreement on how to proceed. Emissions will need to be cut drastically. An average warming of 2 °C (or 1.5 °C) compared to pre-industrial levels is defined as the political threshold of relatively 'safe' warming. At the same time, global emissions tend to either rise or stabilize, showing no sign of the sharp reduction needed to achieve the set political goals (Le Quéré et al. 2018).

In response to this lethargy and its associated anxieties, scientists have started to look at climate engineering, the 'deliberate large-scale manipulation of the planetary environment to counteract anthropogenic climate change', as a possible addition to the climate change portfolio (Royal Society 2009, p. ix). As early as the mid-1960s, climate engineering was being considered as a political and scientific response to anthropogenic climate change (White House 1965). After several cycles of waxing and waning scientific and political interest, climate engineering reemerged as a political option in the mid-2000s, in part due to media attention from an article by Nobel Laureate Paul Crutzen, calling for climate engineering research (Crutzen 2006). Since then, it has consistently moved towards the scientific and political mainstream (Larkin et al. 2018). Just months before the U.S. announced its withdrawal from 'Paris', for example, a group of U.S. American scientists announced their intention to experiment with the injection of calcium carbonate aerosols into the stratosphere, as part of an ongoing

research project into the possibility of artificially cooling global surface temperature by veiling the sun (Burns et al. 2017).

This proposal was the outflow of a broader history of climate modification research conducted in Western academies. Since the 1990s, scientists have predominantly imagined climate engineering in response to anthropogenic climate change, but the search for weather control is intimately connected to historical commercial and military interests (Fleming 2010), and has a long scientific history (Bonneuil and Fressoz 2016). To a significant degree, the development of climate engineering grew out of the military industrial academies of the United States and the Soviet Union (Baskin 2019; Fleming 2010)—its early framing reflecting the epistemologies and ontologies of these academies as well as their systems of measurement (Keith 2000). This longer history has imbued the field with mostly western, historical perceptions of climate and climate change. These determinations of climate change have played a defining role in the development of climate engineering (Fleming 2010). In this essay, I argue that mediating climate change with solar radiation management (SRM), i.e., actively intervening in the Earth's energy budget, entails a specific, holistic view of the earth's climate systems—of the Earth as a global, interconnected whole that can be accurately understood by numerical models. As a result, climate engineering research in general, and solar radiation management (SRM) in particular, primarily engages with quantified, whole-Earth solutions. This view, dependent on a deliberative scientific intentionality that seems connected to anthropocenic notions of responsibility and control, profoundly influences climate engineering's assumptions and research methods. While this may not *necessarily* be to the detriment of climate engineering proposals—in fact, it may be the only workable conception for modifying the global climate—it is a conceptual limit to the enterprise that has to be acknowledged and reckoned with. Additionally, in terms of governance, reliability, and cultural acceptance, this limit could be a fundamental objection to future experimentation (or implementation).

## 2. Wizards and Prophets of Climate Engineering

Climate engineering can roughly be subdivided into two categories; technologies to reduce the carbon concentration in the atmosphere (National Research Council 2015a) and technologies that propose to alter the influx of solar energy into the global climate system (National Research Council 2015b). The wide variety of speculative methods to absorb carbon dioxide back from the atmosphere, often referred to as Negative Emission Technologies (NETs), are increasingly assumed to be an unavoidable part of climate change mitigation (Beck and Mahony 2017; EASAC 2018). Critics and proponents alike have voiced a wide range of concerns about both sets of technologies. These concerns are often specific to particular technologies. Criticism of carbon dioxide removal (CDR) mostly stresses pressures on land-use (Lawrence et al. 2018), doubts about feasibility and effectiveness, and questions of equity and justice (Lenzi 2018). Objections to solar radiation management (SRM), on the other hand, include the unpredictability of intervening in climate systems (Hulme 2014), the potential weaponization of climate control techniques (Barkham 2015), questions about 'whose hands would be on the thermostat' (Nerlich and Jaspal 2012), and the politics inherent to these invasive technologies. Despite these differences, however, both sets of technologies rely on similar, model-driven epistemologies. These methodologies have serious drawbacks. Scholars have argued, for example, that climate engineering is 'undesirable, ungovernable, and unreliable' (Hulme 2014), 'a bad idea' (Robock 2008), undemocratic (Szerszynski et al. 2013), hubristic (Latour 2017), and 'barking mad' (Pierrehumbert 2015)—all criticisms intimately connected to the limitations of the epistemologies underlying climate engineering research. There have, moreover, been multiple attempts to theorize the relationship between humans and their environment, human rights, justice, and possible governance in relation to climate engineering (Emmett and Nye 2017; Preston 2016).

In general, opponents typically criticize climate engineering based on several grounds: its hubristic assumption of being able to control and manage a system as complex as the climate; the unintended consequences that would seemingly inevitably follow climate engineering interventions; questions of justice; and the moral relationship between humans and their environment. Proponents, on the

other hand, tend to insist that climate engineering is the lesser of two evils—a 'bad idea whose time has come' (NPR 2010). As Leslie Paul Thiele observes, in the debate between these two camps, climate engineering is typically seen either as a creative technological option in the case of a climate emergency, or as a hubristic attempt to play God (Thiele 2018). This fits to a wider observation by Mann (2018), who identifies *wizards*—cornucopians who see technological solutions as the key to responsible human existence and sustainability—and *prophets*—neo-Malthusians who argue that humanity (and particularly the West) needs to drastically reduce consumption to sustain themselves on a finite planet. As both Thiele and Mann assert, these positions are difficult (perhaps even impossible) to reconcile. For prophets, the technological optimism of the Prometheans amounts to a dangerous naivety, and an unwillingness to see the drawbacks of technology. For wizards, the rejection of technologies that could save many is 'intellectually dishonest, indifferent to the poor, even racist (because most of the world's hungry are non-Caucasian)' (Mann 2018, p. 7). While Thiele and Mann are correct in observing that these differences are importantly value-loaded, the different outlooks also rely upon a dissimilar reading of epistemology. Duncan McLaren makes this visible by questioning the embedded conceptions of justice in climate engineering modeling, especially for solar radiation management. Observing that these 'emerging technologies would create distinctively new climates, closer to the present climate than those resulting from unabated emissions', but 'with different winners and losers' (McLaren 2018a, p. 209), he argues that there are both explicit and hidden assumptions about justice embedded in the epistemological project around climate engineering. Through a range of model practices, such as the use of extreme counterfactuals and catastrophism, but also through strong claims about certainties, aspects of vulnerability and justice are imbued into the models. Building on work by Bellamy (2015) and Stirling (2003), McLaren asserts that climate engineering 'would appear to involve a wider range and scale of uncertainties than mitigation', which means 'that climate impacts may be more evenly or unevenly distributed than the models imply' (McLaren 2018a, p. 216). Epistemologies, McLaren warns, carry implicit notions of justice and ethics, because certain facets of reality are made visible while others remain obscure.

Flegal and Gupta (2018) make a similar point when they observe that advocates for climate engineering research often frame their argument around a particular conception of 'equity'. According to Flegal and Gupta, these advocates view equity as an empirical question, answerable by scientific analysis, framing equity in terms of winners and losers while simultaneously using this conception of equity to justify climate engineering research. To Flegal and Gupta, this somewhat technocratic view of equity projects, particularly their expert framings of vulnerability and equity, risk excluding the voice of the vulnerable themselves. According to Thilo Wiertz, 'creative play with technological ideas becomes possible through climate modelling', leading these models to 'also become inventive tools, allowing scientists to envision novel ways of climate control and optimization' (Wiertz 2015, p. 438). These observations bring into question how climate engineering researchers know what they know, and what types of questions their epistemology allows them to ask—as well as what the effects of the visions of these scientists are in the development of climate engineering and climate policy. In the remainder of this essay, I reflect upon the global epistemology of climate, its relationship to climate engineering, and how it carries certain normative assumptions.

## 3. The Discovery of the Global: Control and Change

In more ways than one, the scientific understanding of climate change and dreams of climate modification grew up together, sharing the same ideological and epistemological backdrop (Fleming 2010). As Paul Edwards has shown, the scientific understanding of the climate is based upon a constructing mixing models, empirical observations, and historical data into a legible whole (Edwards 2010). One of the most important effects of this process of filtering through computer models, especially combined with satellite observations, is that it enables a *global* view of the Earth's climate system. To make these data legible and intelligible, to make them speak about the phenomena scientists wish them to speak, the data taken from these various sources have to be triangulated against one

another. In Edwards' words, 'virtually everything we now call "global data" is not simply collected; it is checked, filtered, interpreted, and integrated through computer models' (Edwards 2010, p. 188)—a necessary process through which certain aspects and data sets are rendered visible while others remain out of sight.

The development of global data is one of the foremost scientific achievements of the twentieth century. With the development of digital computers during and after WWII, it became possible to model both climate and weather. At first, because of both observational and computational limitations, these models were regional. As the space race commenced, however, satellite imagery and increasing computational capacity made it possible to view the Earth as an interconnected whole, both visually—through the famous Earthrise photograph—and conceptually, through the use of global computer models and representations of a global climate. The importance of this development cannot be overstated. Politically and culturally, it enabled a global 'Spaceship Earth' vision (Höhler 2015). Scientifically, it made it possible to conceptualize a global *climate*. This global view gradually overtook previous (scientific) conceptions that saw weather and climate as predominantly local or regional affairs (Edwards 2010; Hulme 2009). Over the course of the 1970s and 1980s, the new global perspective was translated into new global climate models, strengthening the notion that rising levels of carbon dioxide could influence the climate globally. With an increasingly global vision of the Earth, metaphors of Earth signifying fragility and shared responsibility became more and more prominent. 'Whole Earth' conceptions were solidified in public consciousness by the Earthrise photograph of a blue marble floating in space—and by the use of metaphors such as 'Spaceship Earth', which could function both as a parable of doom and as a call for a shared responsibility for human survival. Although the Earthrise photograph has (rightly) been credited with solidifying this whole Earth conception in the collective imagination, the adoption of a holistic conception of Spaceship Earth had begun to emerge already in the years prior to its capture (Jasanoff 2001). This global view also made a globally changing climate due to human action conceivable. Hart and Victor even go so far as to argue that 'by 1968 the notion that pollution could modify the climate was a commonplace' (Hart and Victor 1993, p. 662).

The global view and the tantalizing prospect of computer-driven weather prediction decisively shaped political and scientific imaginations of climate control. Already in October 1945, a mere two months after the end of WWII, Vladimir Zworykin published the 'outlines of a weather proposal', in which he stressed that 'the importance of accurate, detailed weather prediction, whether regional or worldwide, cannot be exaggerated' (Zworykin 1945, p. 1). Zworykin, a scientist closely linked to the military, saw that the underlying governing physical principles of weather were 'now mostly well-understood'. He argued that prediction, and an increased attempt to understand these processes even better, could conceivably lead to active weather control. Eventually, Zworykin even thought it possible that 'long term climatic changes may be made'. The epistemological leaps that computer models and satellite imagery brought were immediately utilized for militaristic, scientific, and commercial dreams of climate control (Baskin 2019). This immediate adoption of the epistemological tools available to bring the climate under control has an older history too. Many earlier theories about climate and weather combined with intimations of control (Fleming 2010). The dream of climate and weather control was also intimately tied to colonial ideologies of civilizing both the environment and its peoples (Bonneuil and Fressoz 2016; Fleming 2010).

The success of the global view is ubiquitous. The development of a global lens for the weather and the climate was at the very core of the discovery of climate change. And it was also a technopolitical project rooted in Cold War struggles for cultural and technological dominance (Edwards 2010; Hecht and Edwards 2010; Howe 2014). Model projections became relevant because of their capacity to represent global, environmental human impacts. As Edwards notes, 'from the 1990s to the present, a trend toward increasingly comprehensive coupled models of the entire climate system has dominated the field' (Edwards 2011, p. 128). The development of the global view privileges a conversation about the global averaging of climate, about a *global* warming of 1.5 or 2 °C, despite the vast regional differences this entails—not to mention differences in vulnerability (Hulme 2011). Currently, the most

visible emblems of climate change are either local photographs of the destruction of livelihoods and environments, or scientific representations of global phenomena. Climate engineering often follows this logic of a global view of the climate, in which the aim of the technology should be to bring down global temperatures—often an aggregate average.

With the Paris Agreement, it seems, a new phase of anthropogenic climate change politics and the global epistemology of climate has arrived. As of 2015, the Paris Agreement has officially united the world around the goal to keep global warming 'well under 2 °C' and preferably under 1.5 °C—using the global average surface temperature as the measure for climate policy.

## 4. Climate Engineering and the Global View

The reconceptualization of the climate from a regional, typically experiential, phenomenon into a scientized, statistical phenomenon based on global models—making possible new visions of control and management—lies at the heart of current climate engineering research. As its longer history and reliance on global modelling attests, climate engineering is a manifestation of a certain epistemological and ontological relationship between humanity—or, to be more precise: part of humanity—and its environment. As is evidenced by Zworykin, the epistemology of climate and weather science was intimately connected to a notion of control. It is a commonplace observation that the search for control underlay modernity at large (Jasanoff 2003). According to Harari (2014), the start of Western science came from an admission of ignorance, of seeing the natural world as principally knowable but not yet known. To Harari, this admission was part of the larger project of modernity, one that traded meaning for power and control over nature (Harari 2014). The drive to control climate (as a particularly volatile and powerful manifestation of nature) was one major expression. It is manifest in James Espy's attempt at subjugating storms (Espy 1841). It shines through in Ellsworth Huntington's climate determinism and his will to conquer the climate (Huntington 1915). It is clearly visible in the interactions between Svante Arrhenius and Nils Ekholm who speculated about artificially warming the climate 'for the benefit of rapidly propagating mankind' almost immediately after Arrhenius calculated the effect of carbon dioxide as a greenhouse gas (Arrhenius, as quoted in Fleming 1998, p. 74). More recently it is visible in John Martin's proposal to use ocean iron fertilization to capture atmospheric carbon dioxide, summed up in his assertion: 'give me half a tanker of iron and I will give you the next ice age' (Martin, as quoted in Dopyera 1996, p. 28). In these cases, we see that modern theories about how the climate works are often bound up with a drive to bring it under human control.

The first way in which the global epistemology affects climate engineering is by creating a political and social conversation centered on global goals, based on statistically measurable phenomena. Existing political aims, particularly the post-Paris goals of 2 °C (or even 1.5 °C) based predominantly on the IPCC's RCP2.6 scenario, create a discourse around climate change conducive to climate engineering research (Beck and Mahony 2017). Using global average surface temperatures as the main metric for political action, homogenizing regional specificities to a global view, can bring SRM into view as a potential aid. Articles titled 'Solar geoengineering as part of an overall strategy for meeting the 1.5 °C Paris target' can now project scenarios such as 'if solar geoengineering were used to limit global mean temperature to 1.5 °C above preindustrial in an overshoot scenario that would otherwise peak near 3 °C' (MacMartin Douglas et al. 2018, p. 1). The IPCC itself also made mention of SRM in its 2018 special report on the 1.5 °C goal—albeit expressing grave reservations (IPCC 2018). Secondly, the statistical, global vision of the climate makes quantified, whole-Earth solutions appear feasible. Through model ensembles, in which the climatic circumstances of the whole planet can be projected, it becomes possible to speak about restoring 'average surface temperatures by increasing planetary albedo' (Irvine et al. 2019, p. 295). This global perspective evokes specific types of questions. It raises questions about the effectiveness of climate engineering technologies in lowering global surface temperatures, while marginalizing considerations of uncertainty. Additionally, it brings into view model-based discussions about the effect of SRM on regional precipitation, or about the global distribution and

availability of land for the use of carbon capture (and storage) and its uptake potential in the form of BECCS or afforestation.

Awareness of both the benefits and the limitations of this global view may be widespread among climate scientists, but this awareness soon dissipates among other audiences—especially those political communities that shape the global climate goals. One only has to look at the uptake of the 2 °C climate change target to realize this. As Knutti, Rogelj, Sedláček, and Fischer note, this target is somewhat arbitrary. According to them, the '2 °C warming target is perceived by the public as a universally accepted goal, identified by scientists as a safe limit that avoids dangerous climate change. This perception is incorrect: no scientific assessment has clearly justified or defended the 2 °C target as a safe level of warming, and indeed, this is not a problem that science alone can address' (Knutti et al. 2016, p. 13). To Knutti et al., this means that 'global temperature is the best target quantity', but its main use should be to 'anchor discussions', as it is 'unclear what level can be considered safe' (Knutti et al. 2016, p. 13). Still, through its central adoption in the Paris Accord, global mean temperature now legitimizes the SRM debates referenced above—as is evident when MacMartin Douglas et al. (2018) speak about climate engineering as part of a strategy to meet the 1.5 °C Paris goals. There is certainly merit to these views and these model studies. They provide knowledge pivotal to preventing catastrophic climate change, that would otherwise have remained inaccessible. These holistic, model-based visions, however, also fit into a discourse around climate change that privileges scientific expertise over normative discussions.

## 5. Climate Engineering as Entanglement

As Bruno Latour famously observed in the early 1990s, the idea of a separation between humans and their environment was a core belief of modernity (Latour 1991). Climate change, however, brings into clear view both the entanglement of humans and their environment *and* the outsized influence human systems now wield—an influence that could not fully have been grasped with the global epistemology of climate science and its kindred sciences. In so doing, climate change can be seen as a prime manifestation of modernity's obsession with separating nature from culture (Latour 2017). Human influence on the planet is now so vast that the environment and human artifice collide on the largest scales—aided and comprehended through scientific epistemologies that can globalize knowledge. For Paul Crutzen, the scientist who introduced the Anthropocene concept, and Christian Schwägerl, this means that

> *We must change the way we perceive ourselves and our role in the world . . . Rather than representing yet another sign of human hubris, [the Anthropocene] would stress the enormity of humanity's responsibility as stewards of the Earth. It would highlight the immense power of our intellect and our creativity, and the opportunities they offer for the future.*
>
> (Crutzen and Schwägerl 2011)

Through their technological ingenuity and industrial systems, humans have definitely entangled themselves with natural systems on a global scale. In a sense, the Anthropocene is the ultimate Latourian hybrid. With its holistic vision of a homogenous humanity, the Anthropocene is perhaps the quintessential consequence of the global epistemology, of a view based, first and foremost, on *planetary* dimensions, that interweaves human artifice with natural systems (Bonneuil and Fressoz 2016).

The parallels between the Anthropocene and climate engineering can best be seen in the nascent ecomodernist movement, which is a loosely connected movement of scientists, activists, and writers, revolving around rethinking sustainability. Traditionally, 'sustainability is about reversing the negative impacts of contemporary modernity on social equity, human wellbeing and ecological integrity', typically paired with the assertion that 'diverse kinds of technological, environmental and institutional engineering have also interacted with other factors to bring a host of (ostensibly unintended) adverse social and ecological consequences' (Stirling 2019, p. 3). Ecomodernism questions this primary focus. In their own words, ecomodernists 'affirm one long-standing environmental ideal,

that humanity must shrink its impacts on the environment to make more room for nature, while we reject another, that human societies must harmonize with nature to avoid economic and ecological collapse' (Asafu-Adjaye et al. 2015, p. 6). Ecomodernism argues that a retreat from ecological impact is impractical and that, if we, as a global humanity, want to raise global living standards, we have to embrace the power of technology to optimize the way humans have an impact on their surroundings. For ecomodernists the only way to limit destructive human impact is to streamline and modernize production processes—wholeheartedly embracing a Latourian hybridity of human, technological, and natural systems.[1] This redefinition of sustainability, away from reducing the human footprint towards a fully developed entanglement between technology, social processes, and the environment under the guidance of human ingenuity and stewardship, is driven by the holistic epistemologies also prevalent in the climate change and engineering debate. Ecomodernism is an interpretation of the human condition through the scientific lens, aiming to make those epistemologies function in a fairer and more sustainable way. Arguably right about the benefits of the scientific progress made over the past few hundred years, ecomodernism relies on science and technology to capture and improve the human experience and natural environment. The benefit of this view, embodied by the global climate epistemology, is that it makes it possible to create systems of management over human and natural systems, such as SRM technologies and carbon capture technologies. This vision of technological prowess as a means to improve lives and nature carries a notion of anthropocenic responsibility. It is such a techno-optimist outlook that Mann refers when, as quoted earlier, he observes that cornucopians might view the rejection of technological advancements and technological optimism as 'intellectually dishonest, indifferent to the poor, even racist'.

As James Scott famously put it, 'certain forms of knowledge and control require a narrowing of vision' (Scott 1998, p. 11). Such a perceptual restriction brings into 'focus certain limited aspects of an otherwise far more complex and unwieldy reality'. This focus enables a 'high degree of schematic knowledge, control, and manipulation', because it strips away the noise. It limits uncertainties and unpredictability, in favor of a clearly delineated object of knowledge and control. One of the main features of modernity was the search for narrowed, 'predictive models (e.g., risk assessment, cost-benefit analysis, climate modelling) that are designed, on the whole, to facilitate management and control, even in areas of high uncertainty' (Jasanoff 2003, p. 227). Ecomodernism's techno-optimism about the ecological benefits of advances in science and technology (typically in combination with a form of green economic growth), relies on the narrowing of vision of the global epistemology. It strips away many concerns, in order to make possible a measurement of and control over the ecosphere that would benefit humanity as a whole—often with an explicit view at empowering the powerless. It is a similar narrowing of vision that makes it possible to consider climate engineering, particularly SRM. Based precisely on a series of predictive models, predominantly economic and climate modelling, both the Anthropocene and climate engineering tend to narrow visions to a *global whole*. It is this narrowing of vision that introduces the notion of responsibility and stewardship into the Anthropocene debate and the ecomodernist lexicon, portraying humans as part of one whole, as a biological *species* that could control the biosphere. Such a synoptic view of a selective reality is not necessarily problematic. In fact, as Scott (1998) recognizes, it makes possible a high degree of manipulation and control, that would otherwise remain out of reach. The narrowing of vision particular to the climate change debate and climate engineering—and also embedded in the discussion about the Anthropocene—is the universalization of both the human and the global system. It is this universalization that reduces the climate system to manageable metrics, making it possible to imagine SRM as a 'creative technological option' (Thiele 2018), or the capture of carbon dioxide as the solution to climate change. Climate engineering at large, particularly SRM, relies on a global epistemology and a history of climate science that is deeply

---

[1] Although Latour himself would most likely equate this form of hybridity with hubris, with the fallacious assumption that control over nature is possible, as he has done in his criticism of climate engineering in *Facing Gaia.*

connected to this search for control. As Bellamy et al. (2012) recognize, climate engineering proposals tend to 'show a strong emphasis on closed and exclusive 'expert-analytic techniques', closing down 'upon particular sets of problem definition, values, assumptions, and courses of action' (Bellamy et al. 2012, p. 597)[2]. The trouble with this narrowing of vision is that it epistemologically exhibits what Hajer et al.  (2015) refer to as 'cockpitism'—the implicit organizational assumption in which all information flows to a decision-making cockpit. It is on this basis—the essential assumption of a technocratic cockpit embedded within its climate holism—that many have criticized climate engineering and climate policy (Hulme 2009, 2015). Going even further, Erik Swyngedouw has described the global, materialist discourse on climate change as 'the fetishist disavowal of the multiple and complex relations through which environmental changes unfold', particularly through 'the double reductionism to this singular socio-chemical component ($CO_2$)' (Swyngedouw 2010, p. 220). For him, this leads to a state of 'post-politics', which 'is marked by the predominance of a managerial logic in all aspects of life, the reduction of the political to administration where decision-making is increasingly considered to be a question of expert knowledge and not of political position' (Swyngedouw 2010, p. 225). According to McLaren (2016), post-political imaginaries and discourses have framed ethical concerns out of the climate engineering debate—privileging an expert discussion about its desirability and feasibility. Much as earlier climate engineering advocates adopted 'tipping points' as a rhetorical tool appropriated from environmentalist discourse (Heyward and Rayner 2016), the narrative of care and responsibility in climate engineering combines ecomodernism's call for a responsible use of high-technology with an environmentalist critique based on care and repair (McLaren 2018b). In so doing, it creates a sentiment of anthropocenic responsibility for the ecosystem[3], but simultaneously re-inscribes the idea that decision-making is a question of expert knowledge, not an ethical or political position.

## 6. Anthropocenic Limitations to Climate Engineering

The 1991 National Academy of the Sciences report on climate change, which also included a chapter on 'geoengineering' as a possible option, asserted that 'a molecule of $CO_2$ from a cooking fire in Yellowstone or India is subject to the same laws of chemistry and physics in the atmosphere as a molecule from the exhaust pipe of a high performance auto in Indiana or Europe . . . Although the results of climate change will differ from place to place, they derive from global processes' (National Academy of Sciences 1991, p. 3). It is this view that motivates and shapes climate engineering as a research field. The global epistemology dominant in the climate engineering debate homogenizes the climate, disavowing multiple and complex relationships between humans and their environments. It concludes that climate change derives from global processes, and hence can be solved by global interventions. This holistic view of the climate, clearly evident in the rationale of using SRM to bring down global average temperatures, simultaneously homogenizes *and* divides people. It homogenizes through a view of a global climate, to be tweaked at global scales, for a global 'we'. At the same time, it divides by institutionalizing technocratic decision-making and re-inscribing power imbalances. In such post-political, managerial imaginaries of climate change, useful though they may be, much is lost. The deliberative intentionality of altering the climate for the benefit of humankind relies on a 'narrowing' of vision that risks redefining notions of equity, justice, and responsibility in a technocratic way, excluding political debate in the process. The notion of a single humanity as responsible for ecosystems and nature in the Anthropocene, creates a peculiar kind of stewardship. Holistic imaginaries of climate change assume a holistic understanding of the Earth. In order to be able to effectively

---

[2]  Subsequent work by Bellamy et al. (2013) found that using less closed down and expert analytic techniques by using a broader diversity of criteria makes SRM seem like a less desirable and feasible option.

[3]  This rethinking of what 'repair' and care' mean is clearly visible, for example, in the Cambridge Centre for Climate Repair, which researches a variety of climate engineering technologies—such as refreezing the arctic—specifically pointing out the responsibility to 'repair' the climate. The center describes their mission as 'to solve climate change' using climate engineering technology (Ghosh 2019).

intervene, they reduce the vastness of its natural and political reality to manageable levels of complexity, which can then be 'manipulated' using technocratic systems of 'control'.

When it comes to climate engineering, and particularly SRM, an interpretation of the Anthropocene in which humanity acts as a responsible steward holds sway. The catastrophic consequences of the build-up of greenhouses gases in the atmosphere seem all but inevitable—effectively, all that can be influenced now is the scale of the catastrophe—and, so argue the proponents of research, it may be to the benefit of us all—plant, animal and human—to slow warming using our technological prowess. What plagues this conception of SRM, however, is the simplification of climate (and politics) it entails. Global warming is a *hyperobject*, as Timothy Morton describes it (Morton 2013). It cannot be captured wholly, nor is it directly experienced. The only way to create a semblance of capturing global warming as a whole is through a conglomeration of data. Climate engineering seeks to tackle climate change as a holistic phenomenon, a hyperobject made whole, conceptualized as the average or sum total of measurable changes. It relies on meteorological fundamentalism—a term introduced by Meyer (2000) for the assumption, implicitly or explicitly, that the significance of climate can be reduced to its physical characteristics—because it could only be operationalized globally, using global data as an indication of its success. As Swyngedouw (2010) warns, this view can suppress divergent views in favor of techno-managerial planning, solidifying the environmental and social status quo. Couched in the anthropocenic notion of responsibility and stewardship, then, is an implicit project of reframing the ethics of control as an ethics of care and repair—without changing either the global epistemology or the technocratic application. Climate engineering, portrayed as care for humans and their environment, is just one example.

**Funding:** This article is the result of research conducted through the project that has received funding from the European Union's Horizon 2020 research and innovation programme under the Marie Sklodowska-Curie grant agreement No 642935.

**Acknowledgments:** I thank the editors of this special issue and the reviewers for their worthwhile comments and suggestions.

**Conflicts of Interest:** The author declares no conflict of interest.

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
