# Peer review of "Anthropocenic Limitations to Climate Engineering"

_humanities, doi:10.3390/h8040186_

Round 1
Reviewer 1 Report
Thanks for the opportunity to read this interesting paper. I have attached a PDF with more detailed comments on the text, highlighting both points for elaboration and needed typographical corrections. Here I offer some general reflections and suggestions for elaboration/clarification.
As I read it, the paper draws on mixed literatures, including science, history and science and technology studies – to identify some cultural and sociological tendencies in the construction, understanding and research of climate engineering. Confirming preceding scholarship, it highlights relationships between climate engineering, earth systems theory and a scientific anthropocentric sensibility; in which the ‘global view’, modelling projections, potential for systems control, and the construction of a universal humanity combine with the exclusion of non-scientific knowledge to create a narrow ‘synoptic view’ of the prospects for climate engineering.
The paper adds to the existing literature in the way it establishes important and hitherto under-considered connections between ideas of control and the culture and sociology of climate engineering. In conclusion it indicates ways in which climate engineering as a control technology contributes to exacerbating division by institutionalising technocracy and sustains inequality by reinforcing power structures, yet is being rhetorically reconstructed in misleading ways in narratives of repair and care.
While the paper makes sound arguments and draws legitimate conclusions, I would encourage the author(s) to consider elaborating or reinforcing the text in the following areas (in a rough order of importance):
First, to draw out more explicitly the connections (mentioned several times, but not really explained) between anthropocene imaginaries and ‘fear and awe’.
Second, to cite some of the empirical evidence of rhetorical strategies of ‘repair, care, reconstruction’ (eg in the announcement of Cambridge’s ‘Centre for Climate Repair’) to enrich the critique offered.
Third, elaborate the ways in which post-political social imaginaries influence the framings and unde rstandings of climate engineering (see for example McLaren 2016; and Groves 2014).
Fourth, to challenge and critique more strongly (rather than simply reporting) the standard narratives of the history of climate engineering and Anthropocene thinking. Arguably the alleged pivotal role of Crutzen in both these areas is at best an exaggeration. Both have long histories predating his interventions (see for example Bonneuil and Fressoz 2016).
Fifth, (and linked to the fourth) to acknowledge the significance of the cold war not only in the ‘global view’ and modelling practices, but also in climate engineering imaginaries, and more broadly in Anthropocene systems thinking (see for example Bonneuil and Fressoz 2016; and Fleming 2012).
Sixth, sharpen up the critique of ecomodernism, particularly regarding the inconsistencies between technological promises and its rhetoric of equality (as exposed in contrast with the work of Daniel Sarewitz, for example, and also by Flegal & Gupta (2018) in their survey of expert conceptions of equity).
Seventh and finally, to note more explicitly the ways in which the construction of particular understandings of climate engineering in turn leads to the construction of particular socio-technical imaginaries (particularly around BECCS and SAI).
References
Bonneuil and Fressoz 2016, The Shock of the Anthropocene, Verso
Fleming 2012. Fixing the Sky. Columbia University Press.
Flegal and Gupta, Evoking equity as a rationale for solar geoengineering research? Scrutinizing emerging expert visions of equity. Int Environ Agreements (2018) 18: 45. https://doi.org/10.1007/s10784-017-9377-6
Groves, 2014. Care, Uncertainty and Intergenerational Ethics. Palgrave.
McLaren, 2016. Framing out justice: the post-politics of climate engineering discourses. In Preston (Ed) Climate justice and geoengineering: ethics and policy in the atmospheric Anthropocene. Rowman and Littlefield.

Reviewer 2 Report
This paper seeks to argue that the idea of solar radiation management (SRM) geoengineering relies on western, holistic understandings of climate and climate change. It proceeds to divide perspectives on SRM into two camps: ‘wizards’ and ‘prophets’ before describing the emergence of the ‘global view’ to which the author(s) ascribe a causal relationship with SRM. This global view is then critiqued for fostering a ‘narrowing of vision’ that leads to SRM and side-lines alternatives. The paper presents an interesting perspective but I am afraid I cannot yet recommend publication. There are some inaccuracies, some arguments need more evidence to be convincing, and further acknowledgement of and engagement with the existing humanities and social science literature on SRM is required to make clearer its original contribution. I would also encourage the author(s) to proof read the paper once more to tackle a number of missing words, stray letters and repetitions.
Inaccuracies
- p1-2 “American scientists announced their intention to experiment with the injection of aerosols into the stratosphere, with the objective of artificially cooling global surface temperatures by veiling the sun” – this needs to be phrased much more carefully. The scientists (I presume the author(s) mean SCoPEx) do not have the objective of cooling global surface temperature; they are only planning to release less than 2kg of non-toxic calcium carbonate
- p8-9 The characterisation of ecomodernism as “embrac[ing] the power of technology to optimize the way humans have an impact on their surroundings” and being synonymous with “cornucopianism” or “technocracy” seems like a straw man. The ecomodernist manifesto itself stresses “pragmatism” not optimisation and that “Decoupling of human welfare from environmental impacts will require a sustained commitment to technological progress and the continuing evolution of social, economic, and political institutions alongside those changes”. At the very least this makes ecomodernism a broad church. Are the author(s) implying that the inclusion of any technology at all in a vision constitutes cornucopianism and technocracy? The author(s) need to either significantly strengthen their characterisation or drop it. I suggest the latter: why not simply focus on ‘those who advocate for SRM’?
- p10 “The holistic view of the climate, clearly evident in most SRM proposals” – while some SRM approaches involve an inherently holistic ‘global’ implementation (e.g. stratospheric aerosols, space reflectors) I would suggest that most actually do not. Urban, desert or crop albedo modifications, ocean microbubbles and marine cloud brightening are all implemented locally. If you stretch the meaning by saying that these too have global effects then any kind of emissions reduction (e.g. renewable energy technologies, behavioural changes) is also ‘holistic’ by the same logic. I suggest the author(s) delimit the scope of their discussion to transboundary SRM or something similar, and reflect that in the title.
Evidence
p2 and 4 “intimately connected to historical commercial and military interests” – more examples of these alleged commercial and military interests are needed. It is not really demonstrated in the paper that the relationship is ‘intimate’
p4 “earlier theories about climate and weather were combined with intimations of control and the dream of climate and weather control was intimately tied to colonial ideologies” – how exactly were these intimately tied to colonial ideologies?
p5 “This admission, moreover, was part of a larger project of modernity, one that traded meaning—a mystical world brimming with meaning—for power and control over nature” – if science and understanding = power and control then that power and control can be used for good as well as bad. I get the sense here and throughout that the author(s) see only the bad. The author(s) need to be explicit about specifically what other forms of knowledge they mean and what practical value they add – this would also strengthen the conclusion, which suggests alternatives but does not go into any detail.
Wider literature
p3 the discussion around assumptions in models would be strengthened by way of engagement with Flegal and Gupta (2017)
p1 and 9 the narrowing of visions (including in models) and the risks of this seems to be a main thread running through the paper - this point should engage with Bellamy (2012) who also cautioned against narrow technical frames leading to particular geoengineering approaches, and his subsequent work on opening up to alternatives (2013)
p10 “reframing the ethics of control as an ethics of care and repair—without changing either the global epistemology or the technocratic application” – this strikes me as a case of ‘stolen rhetoric’ where one cultural position steals the language of another in order to justify an approach normally objectionable to that culture. The author(s) should engage with Heyward and Rayner (2016)
Minor points
p1 “Climate engineering, imagined as a technological reaction to possible climate emergencies” – this is not the only way SRM has been justified. In fact, it has lost a lot of traction to due heavy criticism from the social sciences and humanities (e.g. Sillman (2015), Markusson (2013)
p1 “increasing numbers of scientists have started to look at climate engineering” – my sense is that research interest in SRM has actually decreased if anything, and it always seems to be the same old scientists doing the research
P1 “as an alternative solution” – I’m not aware of a credible actor in this field who has framed it as an alternative to mitigation
References
Flegal and Gupta (2017) Evoking equity as a rationale for solar geoengineering research? Scrutinizing emerging expert visions of equity. International Environmental Agreements: Politics, Law and Economics
Bellamy et al. (2012) A review of climate geoengineering appraisals. WIREs Climate Change
Bellamy et al. (2013) ‘Opening up’ geoengineering appraisal: Multi-Criteria Mapping of options for tackling climate change. Global Environmental Change
Heyward and Rayner (2016) Apocalypse nicked! Stolen rhetoric in early geoengineering. In: Anthropology and Climate Change: From Actions to Transformations
Sillman et al. (2016) Climate emergencies do not justify engineering the climate. Nature Climate Change
Markusson et al. (2013) ‘In case of emergency press here’: framing geoengineering as a response to dangerous climate change. WIREs Climate Change
Round 2
Reviewer 2 Report
I would like to thank the author(s) for their most thorough and considered responses to my comments. The article is significantly improved and I am pleased to recommend the article for publication with no further revisions.
Author Response
My thanks again to the reviewer for their worthwhile comments, and for re-engaging with the text.